# Expression of the Z Variant of α1-Antitrypsin Suppresses Hepatic Cholesterol Biosynthesis in Transgenic Zebrafish

**DOI:** 10.3390/ijms24032475

**Published:** 2023-01-27

**Authors:** Connie Fung, Brendan Wilding, Ralf B. Schittenhelm, Robert J. Bryson-Richardson, Phillip I. Bird

**Affiliations:** 1Department of Biochemistry and Molecular Biology, Biomedicine Discovery Institute, Monash University, Melbourne 3800, Australia; 2Monash Proteomics and Metabolomics Facility, Monash University, Melbourne 3800, Australia; 3School of Biological Sciences, Monash University, Melbourne 3800, Australia

**Keywords:** α-1-antitrypsin, SERPINA1, zebrafish, liver, ERAD

## Abstract

Individuals homozygous for the Pi*Z allele of SERPINA1 (ZAAT) are susceptible to lung disease due to insufficient α1-antitrypsin secretion into the circulation and may develop liver disease due to compromised protein folding that leads to inclusion body formation in the endoplasmic reticulum (ER) of hepatocytes. Transgenic zebrafish expressing human ZAAT show no signs of hepatic accumulation despite displaying serum insufficiency, suggesting the defect in ZAAT secretion occurs independently of its tendency to form inclusion bodies. In this study, proteomic, transcriptomic, and biochemical analysis provided evidence of suppressed Srebp2-mediated cholesterol biosynthesis in the liver of ZAAT-expressing zebrafish. To investigate the basis for this perturbation, CRISPR/Cas9 gene editing was used to manipulate ER protein quality control factors. Mutation of *erlec1* resulted in a further suppression in the cholesterol biosynthesis pathway, confirming a role for this ER lectin in targeting misfolded ZAAT for ER-associated degradation (ERAD). Mutation of the two ER mannosidase homologs enhanced ZAAT secretion without inducing hepatic accumulation. These insights into hepatic ZAAT processing suggest potential therapeutic targets to improve secretion and alleviate serum insufficiency in this form of the α1-antitrypsin disease.

## 1. Introduction

α1-antitrypsin (encoded by *SERPINA1*) is an acute-phase serum protein that regulates neutrophil elastase, primarily in the lungs. It is synthesised and secreted by hepatocytes [1]. In α1-antitrypsin deficiency (AATD), low serum levels of α1-antitrypsin predispose affected individuals to neutrophil elastase-mediated lung damage, leading to the development of emphysema, bronchiectasis, and other chronic obstructive pulmonary disease [2]. While a number of mutations in *SERPINA1* contribute to respiratory disorders by altering mRNA processing which results in the complete absence of circulating α1-antitrypsin (null alleles) [3], some mutations are known to cause defects in protein folding, resulting in the retention of the misfolded protein in the endoplasmic reticulum (ER) and secretion insufficiency [4,5,6,7,8,9,10,11,12,13]. The Z-allele (E342K) is commonly found in patients with severe AATD, with circulating α1-antitrypsin levels in ZZ homozygotes (PiZZ) being only 10–15% of normal individuals [14].

Z-antitrypsin (ZAAT) has the propensity to polymerise, resulting in the deposition of glycoprotein aggregates in the ER of hepatocytes which can be detected histologically as Periodic acid Schiff-positive, diastase-resistant, inclusion bodies. The number of inclusion bodies correlates with hepatic stress and the progression of liver fibrosis, suggesting ZAAT aggregates are hepatotoxic [15,16]. The development of severe liver disease in PiZZ individuals generally occurs in two distinct age groups. An ongoing Swedish longitudinal study has revealed that 10% of Pi*ZZ individuals develop cholestatic jaundice and liver cirrhosis during childhood [17], while the liver function is normal for the remaining cohort through to adulthood [18,19]. Liver cirrhosis is also commonly observed in Pi*ZZ individuals over 50 years of age [20]. The reason(s) for this variability in onset and symptom severity is unknown but suggests that the majority of Pi*ZZ patients have mechanisms to cope with aggregating ZAAT protein in the liver. 

Evidence of both proteasomal and lysosomal-mediated mechanisms in the clearance of ZAAT in hepatocytes indicates that multiple systems work to reduce stress elicited by misfolded ZAAT in the ER [21,22,23,24,25,26,27]. It is generally accepted that soluble misfolded ZAAT monomers in the ER are targeted to the proteasome via the process of ER-associated degradation (ERAD), while insoluble polymers are removed by autophagy or via ER-to-lysosome-associated degradation [28,29]. Exhaustion or defects in these responses may therefore lead to the development of symptomatic liver diseases.

We have previously described a transgenic zebrafish model for ZAAT-mediated α1-antitrypsin deficiency [30]. A unique feature of this model is the absence of ZAAT inclusion bodies in hepatocytes (Appendix A), although the markedly reduced serum levels of ZAAT seen in Pi*ZZ individuals are evident. It appears that zebrafish possess highly efficient proteostasis networks that clear misfolded ZAAT, preventing aggregation in hepatocytes. This model also indicates that the serum deficiency in mammalian systems is a result of inefficient folding and release of the variant protein, and is not simply due to hepatic ZAAT sequestration lowering the amount of material available for secretion.

In this study, we carried out transcriptomic and proteomic analyses of zebrafish liver to identify systems involved in ZAAT clearance. This revealed perturbation of the cholesterol biosynthesis pathway in response to ZAAT expression which was confirmed by biochemical assays. Genetic manipulation of ERAD system components regulating both cholesterol metabolism and clearance of misfolding ZAAT provided insights into the interrelationship of these processes in hepatocytes.

## 2. Results

### 2.1. Expression of ZAAT in Zebrafish Liver Results in Suppression of Cholesterol Biosynthesis

Extensive studies in mammalian systems indicate that activation of the ERAD and autophagy [24,25,26,31,32,33,34,35] but not the unfolded protein response (UPR) pathways are associated with ZAAT accumulation and clearance in hepatocytes [36,37,38]. To identify pathways activated and operating in our zebrafish model, RNA-seq and LC-MS/MS experiments were carried out in livers harvested from two-month-old adult fish (*n* = 6). Four zebrafish genotypes were analysed: (1) those expressing human wildtype AAT (A+); (2) AAT non-transgenic siblings (A-); (3) those expressing human mutant ZAAT (Z+); and (4) ZAAT non-transgenic siblings (Z-). Each liver was divided for LC-MS/MS and RNA-seq analysis so that protein and mRNA levels could be correlated within each individual to capture changes occurring at both the transcript and protein levels (Figure 1A).

RNA-seq output data were aligned to a modified Ensembl zebrafish genome reference sequence containing the AAT transgene using RNAsik and STAR [39,40]. >95% of filtered reads aligned to the genome and >60% aligned to a feature, identifying 28,210 unique genes. The RNA-seq data showed that the transgene expression levels (driven by the *fabp10a* promoter) fall within the top 0.2–1.0% of all genes expressed (Appendix A), similar to the levels of endogenous SerpinA1 genes.

Approximately 30,000 peptides were detected from LC-MS/MS experiment, resulting in quantitative data for 4420 unique proteins based on a data-independent acquisition (DIA) approach using a pre-existing spectral library (Figure 1B). Approximately 94% of uniquely mapped proteins from the LC-MS/MS also appeared in the RNA-seq dataset, however, the transcript and protein levels correlated poorly (Person’s *r* = 0.2175) (Figure 1C). This highlights the importance of using both techniques in parallel to explore pathway dysregulation in our disease model.

Multiple Student’s *t*-tests with summed false discovery rate (FDR) threshold were used to compare the transcript/peptide levels between the Z+ and other three genotype groups (Appendix A). Any transcript/peptide with significant dysregulation in the Z+ genotype was identified as a differentially expressed gene/protein (DEG/DEP). In the RNA-seq analysis, 314 up-regulated and 121 downregulated genes were identified in the Z+ liver using this approach (Appendix A). In the LC-MS/MS analysis, conditions with lower stringency were used due to low sample numbers in the non-transgenic groups. 48 proteins were found to be up-regulated and 29 proteins were downregulated in the Z+ liver (Appendix A). The lists of DEGs and DEPs identified do not overlap with each other, suggesting significant post-transcriptional regulation of the DEP genes (Figure 1D).

The biological roles of the DEGs/DEPs were assessed using two well-established databases for functional annotation, the Kyoto Encyclopedia of Genes and Genomes (KEGG) and the Gene Ontology (GO) consortium, due to their extensive curation and applicability to a wide range of species. To identify any perturbed pathways in Z+ livers, an in-house pipeline using the GO consortium website and KEGG mapper to map functional annotations to DEGs/DEPs. For the RNA-seq data set, 377 DEGs (86.7%) were successfully mapped to GO terms, and 166 genes (38.2%) were mapped to KEGG pathways. For the LC-MS/MS data set, 68 (88.3%) and 39 (50.6%) DEPs were mapped to GO and KEGG pathways respectively (Figure 1D).

Functional enrichment analysis was then performed to determine whether any KEGG pathways or GO terms are over-represented in the DEG/DEP lists. An enrichment score of 1 indicates that a gene/protein from the DEG/DEP lists that have been mapped to a functional annotation occurs at a similar frequency to random DEGs/DEPs. Annotations with an enrichment > 1 were considered to be over-represented, suggesting perturbation of these processes in the ZAAT-expressing liver. Ingenuity pathway analysis software (QIAGEN Inc., https://digitalinsights.qiagen.com/IPA) was used to confirm the findings and predict upstream regulators of dysregulated pathways [41]. These were investigated further.

GO terms under the ‘biological process’ category were used for the identification of dysregulated pathways in our data sets. 431 GO terms were identified as over-represented in the RNA-seq data set (DEGs) (Appendix A), with annotations involving cholesterol biosynthesis being the most obvious. The reduced transcript levels for DEGs mapping to cholesterol biosynthesis GO terms suggested this pathway is suppressed in the Z+ liver (Figure 2). Although genes encoding several components of the autophagy-initiation complex were identified as DEGs, consistent with the role of autophagy in ZAAT disposal reported in mammalian models [26,33,34,35,42,43], significant up-regulation of other key factors involved in nucleation and maturation of the autophagosome was not observed. Furthermore, PI3P phosphatases with antagonistic roles in autophagy were also identified as up-regulated DEGs. This suggests that ZAAT expression does not drive a systemic autophagic response in zebrafish liver.

A total of 68 GO terms and 23 KEGG pathways were over-represented in the LC-MS/MS data set (DEPs) (Appendix A), with the most enriched pathway from KEGG mapping being “protein processing in endoplasmic reticulum” (Figure 3). This pathway is also enriched in the proteome of human Pi*ZZ hepatocytes [44]. Up-regulation in the expression of Erlec1, Vcp, Ufd1 and several proteasome subunits indicated activation of the ER quality control pathway (ERAD). Interestingly, annotations associated with autophagy were also enriched in the LC-MS/MS data (i.e., the DEPs; Vcp, Rab1aa and Rab1ab). However, these proteins are also known to support other cellular processes, such as ubiquitination-dependent processes and vesicle transport. When considered with the RNA-seq analysis, we concluded from our analysis that there is no convincing evidence for macroautophagy activation in the Z+ zebrafish liver.

The KEGG pathway ‘Cholesterol metabolism’ was enriched in the LC-MS/MS data set, with increased levels of the transport/uptake factors APOB and LRP1 detected in Z+ samples. This suggests the activation of compensating mechanisms to maintain cholesterol homeostasis, countering the reduced cholesterol synthesis in the liver indicated by the RNA-seq analysis.

Quantitative PCR (qPCR) was used for the validation of RNA-seq data and subsequent interrogation of the cholesterol biosynthesis pathway. To ensure that the qPCR and RNA-seq approaches yielded consistent results, six DEGs were selected for qPCR validation comparing liver RNA from Z+ fish to A+ controls (Figure 4C). These comprised three significantly downregulated genes (*mvk*, *fdps,* and *lss*) and three up-regulated genes (*atg101*, *map1lc3a*, and *bnip4*). The Log2 fold change of the Z+: A+ ratio in expression levels obtained from qPCR using a different set of fish was compared to those obtained from the RNA-seq data. The expression pattern of selected genes from the qPCR experiment correlated well with RNA-seq data (Spearman correlation *r* = 0.9429), thus validating the RNA-seq analysis and supporting the use of qPCR for further interrogation of DEGs.

### 2.2. Cholesterol Biosynthesis Suppression Is a ZAAT-Specific Response in Both Zebrafish and Mammalian Models

Detailed analysis of the cholesterol biosynthesis pathway revealed that genes encoding 12 out of the 21 enzymes involved in the stepwise conversion of acetyl-CoA to cholesterol were identified as DEGs in our datasets, with significant reductions in transcript levels (2 to 7-fold) in Z+ livers (Figure 4A). Consistently, the top five canonical pathways in the RNA-seq data identified by the Ingenuity pathway analysis are involved in cholesterol biosynthesis. The software also predicted the inhibition of the transcriptional factor SREBP2 (encoded by *SREBF2*) as an upstream regulator to explain the suppression of cholesterol biosynthesis (Appendix A). 

The lower transcripts levels of multiple SREBP2-regulated cholesterol pathway genes suggested that the hepatic cholesterol levels would be lower in Z+ fish due to reduced biosynthesis. Total cholesterol was measured in adult zebrafish livers. The results showed a significant reduction of hepatic cholesterol in the Z+ samples compared to the three other genotypes (Figure 4B). This directly demonstrated that expression of ZAAT results in the lowering of hepatic cholesterol at a physiological level, and is consistent with an effect on SREBP2-mediated cholesterol biosynthesis. 

To determine whether the suppression of cholesterol biosynthesis as a response to ZAAT expression is restricted to the zebrafish system, or if it also potentially occurs in α1-antitrypsin deficiency patients, the expression of SREBP2-regulated cholesterol pathway genes was compared via qPCR in human hepatic HepG2 cells stably expressing either eGFP-T2A-AAT or eGFP-T2A-ZAAT (Figure 5). The use of the T2A motif allows separate but stoichiometric expression of eGFP and antitrypsin from the same transcriptional unit. eGFP provided the means to compare expression levels of transfected AAT and ZAAT, as antitrypsin produced from the transfected plasmids cannot be distinguished from endogenous antitrypsin in HepG2 cells. Immunoblotting analysis of eGFP indicated that expression of the AAT transgene was higher than ZAAT in transfected cells (Figure 5A). 

Five key genes involved in the cholesterol biosynthetic pathway were assessed in HepG2 cells. The results showed a reduction in the expression of *HMGCS*, *MVK*, *FDPS* and *LSS,* and no change in *HMGCR* in cells expressing ZAAT, consistent with the zebrafish RNA-seq analysis (Figure 5B). By contrast, no decrease in the expression of these genes was observed when comparing cells expressing AAT to non-transfected cells, despite the much higher expression level of AAT (Figure 5C). These results indicate that, as in the zebrafish system, the SREBP2-mediated cholesterol biosynthesis pathway is suppressed in ZAAT-expressing human cells. 

### 2.3. Compromising the HRD1 E3 Ligase Complex Further Suppresses Cholesterol Biosynthesis but Does Not Influence ZAAT Accumulation or Secretion

As shown above, RNA analysis in both zebrafish and human models strongly suggested SREBP2 activation is suppressed in ZAAT-expressing hepatocytes. SREBP2-mediated signalling is tightly controlled in hepatocytes to ensure cholesterol homeostasis [45]. The precursor complex (SCAP-SREBP2) is regulated by the sterol-dependent binding of SCAP to the short-lived anchor proteins Insig1 or Insig2 in the ER. The turnover of Insig1 in particular via ERAD influences the extent of SCAP-SREBP2 interaction with the COPII components required for translocation to the Golgi and activation of *de novo* cholesterol biosynthesis [46,47]. 

Two mammalian homologs of the yeast Hrd1 E3 ligase, HRD1 and gp78, have been reported to facilitate the clearance of ZAAT via ERAD [24,25]. As gp78 (but not HRD1) is also involved in the turnover of Insig proteins [47], we predicted that the burden of processing misfolded ZAAT in the zebrafish liver limits the capacity of the gp78 E3 ligase complex to correctly regulate Insig1 levels, resulting in increased retention of SCAP-SREBP2 complex in the ER and preventing the activation of the cholesterol biosynthesis pathway. 

The simplest way to test this hypothesis would be to increase the competition between ZAAT and Insig1 for the gp78 ligase. This can be achieved by targeting the Hrd1 E3 ligase so that misfolded ZAAT would be redirected to the gp78 ligase complex for ER protein turnover. If the hypothesis is true, we should expect further Insig1 stabilisation and suppression of cholesterol biosynthesis. However, targeting the Hrd1 complex in vivo cannot be achieved as deletion of integral components of the HRD1 E3 ligase complex including HRD1, SEL1L, and Derlin-1 are embryonically lethal [48,49,50]. We, therefore, propose that a reduction (but not complete ablation) of Hrd1 complex activity may be sufficient to redirect more ZAAT to the gp78 complex, without killing the cell.

The ER lectins, Erlec1 and Os-9, are luminal components of the Hrd1 complex responsible for recruiting misfolded substrates including antitrypsin to the E3 ligase and are reported to have partially redundant roles [51,52,53,54,55]. Therefore, targeting one of these lectins should reduce ZAAT degradation via the Hrd1 complex without causing lethality. Since our LC-MS/MS analysis identified zebrafish Erlec1 as a DEP that is up-regulated in the Z+ fish liver, Erlec1 was chosen over Os-9 for genetic manipulation. The CRISPR/Cas9 gene editing system was used to generate an *erlec1* mutant line (Appendix A).

Reverse-transcriptase PCR (RT-PCR) analysis showed a pronounced reduction of *erlec1* transcripts in the homozygous mutant animals (Figure 6A), consistent with nonsense-mediated RNA decay. Any transcripts escaping nonsense-mediated decay would produce a very small, truncated protein (Appendix A), and therefore no functional Erlec1 is likely to be produced from this allele. 

In human embryonic kidney cells (HEK293), knockout of both ERAD lectins ERLEC1 and OS-9, but not single knockouts, results in instability of the HRD1 complex [55]. To determine if the knockout of *erlec1* alone is sufficient to hinder Hrd1-mediated processing in zebrafish, the transcripts of key components such as Hrd1 (also known as Syvn1), Sel1l and Os-9 were compared between *erlec1*^−/−^ and wildtype livers in adult fish via RT-PCR. Reduction in expression of these components was observed in *erlec1*^−/−^ animals (Figure 6B), suggesting the Hrd1 complex is likely to be compromised.

The expression of Srebp2-regulated cholesterol pathway genes was then assessed via qPCR. As expected, the results revealed no significant difference in the expression levels of these genes between wildtype and *erlec1*^−/−^ livers in the absence of ZAAT (Figure 6C) because gp78 remains functional and the turnover of Insig does not depend on the HRD1 complex. By contrast, in livers expressing ZAAT, expression of the cholesterol genes was significantly decreased in *erlec1*^−/−^ fish compared to wild-type siblings (Figure 6D). This supports our hypothesis that the removal of Erlec1 alters the ER processing of misfolded ZAAT in zebrafish liver in a way that further suppresses Srebp2 activation. This is most likely due to increased competition between ZAAT and Insig1 for gp78 because the Hrd1 complex is compromised. 

Next, we asked whether the further suppression of cholesterol biosynthesis genes in Z+, *erlec1*^−/−^ fish results in a further reduction of hepatic cholesterol. Total cholesterol in fish expressing ZAAT was measured. However, no difference in total cholesterol was detected in the Z+ *erlec1*^−/−^ liver, compared to Z+ *erlec1*^+/+^ siblings (Figure 6E). As total cholesterol is reduced in Z+ liver without manipulation of Erlec1, these results suggest that further suppression in cholesterol biosynthesis in the absence of Erlec1 may be opposed by compensatory mechanisms that maintain minimum tissue cholesterol levels, such as an increase in uptake of cholesterol from the circulation.

Suppression of Hrd1-mediated processing may reduce the amount of ZAAT exiting the ER for degradation and thus promote aggregation of polymerogenic ZAAT. To determine if the loss of Erlec1 affects ER retention of ZAAT, we checked for the presence of inclusion bodies in the Z+ *erlec1*^−/−^ liver. Inclusion bodies in Pi*ZZ patients (and PiZ transgenic mice) are routinely detected using histological staining of liver sections with Periodic acid Schiff (PAS) solution, indicating the presence of glycogen and mucosubstances. The addition of diastase during staining effectively removes glycogen, so any inclusions that remain positively stained in hepatocytes indicate the aggregation of ZAAT glycoproteins. Paraffin-embedded sections of adult Z+ zebrafish livers were stained with PAS and diastase and showed no evidence of inclusion bodies in either *erlec1*^+/+^ or *erlec1*^−/−^ animals (Figure 6F), consistent with liver sections stained for human antitrypsin (Appendix A).

To investigate the effect of *erlec1*^−/−^ on ZAAT secretion, ZAAT was detected in blood by immunoblotting. Liver extracts from the same animals were subjected to immunoblotting for eGFP to assess transgene expression level. Quantification of protein signals revealed no difference in the level of circulating ZAAT in the *erlec1*^−/−^ fish compared to *erlec1*^+/+^ siblings (Figure 6G). Taken together with the absence of inclusion bodies in the *erlec1*^−/−^ liver, our results suggest the clearance of ZAAT in the liver remains efficient despite compromised Hrd1-mediated degradation.

### 2.4. Deletion of Man1b1 Mannosidases Increases ZAAT Secretion without Inducing Intracellular Accumulation

Our data from the *erlec1*^−/−^ fish indicated that increasing the ERAD burden by processing ZAAT enhances suppression of Srebp2 activation in the zebrafish liver. We, therefore, postulate that Srebp2 signalling suppression might be released if ZAAT is directed away from the ERAD system. The extraction of misfolded proteins from the folding cycle and sorting to ERAD involves the modification of N-glycans by ER glucosidases and mannosidases [31,32]. Kifunensine, an inhibitor of ER mannosidase I (encoded by *MAN1B1*), delays the degradation of ZAAT and increases its secretion, presumably by prolonging the time available for slow folding intermediates in the folding cycle to gain secretion competency by allowing them to escape ERAD. Based on these observations, mutation of MAN1B1 homologs in ZAAT-expressing zebrafish liver was predicted to reduce the amount of ZAAT delivered to the ERAD systems, thus relieving pressure on gp78 and rescuing suppression of the cholesterol biosynthetic pathway. We also predicted an increase in circulating ZAAT because in the absence of ERAD, slowly folding or misfolding molecules may eventually attain secretion competency.

There are two MAN1B1 homologs in zebrafish (Appendix A). Double mutant animals were generated by simultaneous injection of CRISPR gRNAs targeting both MAN1B1 homologs (Figure 7A). RT-PCR analysis of *man1b1a* confirmed the loss of transcript in mutant animals presumably due to nonsense-mediated decay (Figure 7B). Nonsense-mediated decay of the mutant *man1b1b* transcript did not occur, although an increase in molecular mass of the mutant mRNA is indicative of a +47 bp insertion detected by sequencing (Appendix A). As the mutation disrupts the reading frame of *man1b1b*, functional Man1b1 proteins will not be produced in the double mutant fish.

To examine the effect of the loss of Man1b1 on ZAAT secretion, circulating levels of ZAAT were analysed by immunoblotting. No differences in circulating levels of ZAAT were detected between wildtype and fish lacking either Man1b1a or Man1b1b (Appendix A), but a significant increase in circulating ZAAT was observed in *man1b1a*^−/−^
*man1b1b*^−/−^ fish (Figure 7C). The increase in circulating ZAAT was not due to higher ZAAT transgene expression in the *man1b1a*^−/−^
*man1b1b*^−/−^ fish based on eGFP expression in the liver. These results suggested that the two zebrafish Man1b1 homologs share redundant roles, and that removal of both homologs increases ZAAT secretion, which is consistent with the results of chemical targeting of MAN1B1 in human cell line studies. 

Although treatment with kifunensine in human cells boosts ZAAT secretion, intracellular retention of ZAAT also increases due to reduced degradation [31,32]. Therefore, we asked whether the increased ZAAT secretion observed in the *man1b1a^−^*^/−^
*man1b1b*^−/−^ fish is accompanied by accumulation in hepatocytes. However, staining of adult fish livers for human antitrypsin or PAS with diastase digestion revealed no inclusion bodies (Figure 7D, Appendix A), suggesting the misfolded ZAAT does not accumulate in hepatocytes upon Man1b1 deletion.

The lack of hepatic ZAAT accumulation in the absence of functional Man1b1 proteins was perplexing, as Man1b1 is generally regarded as an integral part of the ER quality control system. To determine whether the loss of Man1b1 alleviates the processing pressure on gp78, the cholesterol biosynthetic pathway was monitored via qPCR. No significant change was observed in the expression levels of cholesterol biosynthetic genes in either single- or double-Man1b1 mutant livers compared to wildtype, in either the non-transgenic or ZAAT-expressing background (Figure 7E). These results indicated that the removal of Man1b1 proteins does not ease the burden on the gp78 ligase complex, and that processing of Insig1 and Srebp2 activation remains limited in the Z+ liver. This suggests that the targeting of ZAAT to ERAD ligase complexes is not dependent on Man1b1 and is supported by our histological observations that ZAAT does not accumulate in the *man1b1a^−^*^/−^
*man1b1b*^−/−^ fish liver.

## 3. Discussion

In our transgenic zebrafish model, ZAAT does not accumulate in hepatocyte inclusion bodies but the secretion insufficiency observed in human Pi*ZZ individuals is evident [30]. 

The RNA-seq data described here show that both AAT and ZAAT transgenes are expressed at comparable levels as endogenous antitrypsin. This rules out the trivial explanation that the level of ZAAT expression is insufficient in the zebrafish model to drive the formation of inclusion bodies. It further strengthens our prediction that the lack of hepatic ZAAT accumulation in this model is due to the efficient removal of the misfolded protein.

It is generally accepted that autophagy is induced by ZAAT accumulation in the human and mouse liver, and is involved in its clearance [26,33,34,35]. The lack of hepatic ZAAT accumulation in the transgenic zebrafish liver may explain why autophagy is apparently not activated in our model. Despite functional enrichment analysis of some autophagy genes in the RNA-seq data, detailed inspection of these DEGs revealed increases in both inducing and repressing autophagy factors in the Z+ liver, with no indication of a general activation of autophagy. 

Autophagy components that were induced have documented roles in other pathways. For example, ULK proteins have autophagy-independent roles such as regulating ER to Golgi vesicular trafficking, redox homeostasis, and cholesterol metabolism [56,57,58]. In particular, ULK1 promotes AKT/FOXO3a-mediated *SREBP2* expression [57]. Therefore, up-regulation of ULK1/2 homologs in our zebrafish model may present a mechanism opposing ZAAT-induced stress by increasing ER cargo export to alleviate an accumulated protein load, or it may be a specific feedback response to low cholesterol levels that expedites the movement of Srepb2 to the Golgi via COPII vesicles, or increases *srebp2* expression. In future studies, biochemical analysis on autophagy flux in this model would provide a definitive indication of whether autophagy is indeed activated in the Z+ liver. 

The most striking result from our bioinformatic analysis of Z+ liver is the suppression of SREBP2-regulated cholesterol biosynthesis genes, which we demonstrated biochemically results in lower hepatic cholesterol, and also occurs in human liver cells expressing ZAAT. This is also consistent with microarray analysis results in a transgenic mouse model with doxycycline-inducible expression of ZAAT (referred to as the inducible Z mouse) [59], as well as RNA-seq analysis in the PiZ mouse model [60]. However, observations in the inducible Z mouse have not been followed up to determine if hepatic cholesterol levels are actually affected. Interestingly, the hepatic cholesterol level in PiZ mice was reported to be significantly higher compared to PiM controls, contrasting the RNA-seq data from the same study. Taken together, the effect of ZAAT expression on cholesterol metabolism in the mouse models remains unclear. 

In another serpinopathy, inclusion bodies composed of polymerised neuroserpin (SERPINN1) proteins are thought to have gain-of-function toxicity in the brain, resulting in familial encephalopathy, a rare form of dementia [61]. Expression and accumulation of polymerised neuroserpin activates the ER overload response but not the UPR and is associated with up-regulation of the SREBP2-dependent cholesterol biosynthetic pathway [62]. 

Acknowledging the extraordinary complexity of the circuits and components that regulate cholesterol metabolism, which includes at least four major regulators (SREBP2, SCAP, Insig, HMGCR) controlled by various E3 ubiquitin ligases including gp78, RNF145 and HRD1 [63,64], the simplest way to reconcile the apparently contradictory impacts of ZAAT and polymerized neuroserpin on cholesterol genes is to posit that they interact differentially with the key ERAD ligases. This is supported by the fact that HMGCR, which is primarily regulated by gp78/RNF145, was not identified as a DEG in our system, but is up-regulated in the presence of polymerized neuroserpin [62]. Our results suggest that ZAAT in zebrafish hepatocytes is handled by both Hrd1 and gp78, and its lack of intracellular accumulation indicates that although they are perturbed these systems are not overloaded. By contrast, polymerized neuroserpin is primarily handled by gp78 [62], and its intracellular accumulation indicates that the ligases are overloaded. Thus, given that Insig1 drives degradation of HMGCR via gp78/RNF145, competition by neuroserpin stabilizes Insig1/HMGCR thus increasing cholesterol production. 

The correlation between mutant serpin accumulation and increased cholesterol biosynthesis reported in the neuroserpin model [62] is also consistent with the steatosis phenotype with elevated levels of hepatic cholesterol in the PiZ mouse liver [60]. Taking these studies together, and remembering that cholesterol gene suppression occurs in inducible PiZ mouse liver [59], zebrafish hepatocytes and HepG2 cells (reported here) it might be suggested that early in liver disease progression, before inclusion bodies form, perturbation of the E3 ligases favours partial Insig1 stabilization, SREBP2 suppression and lowered cholesterol synthesis. As the disease progresses and inclusion bodies form, the ligases become overloaded, Insig1 is fully stabilized, HMGCR levels build up and cholesterol synthesis increases even if SREBP2 levels are low. Of course, we do not rule out the induction of other cellular stress responses affecting cholesterol metabolism. 

To our knowledge, we are the first to propose a molecular mechanism for cholesterol biosynthesis suppression in the presence of ZAAT: that increased processing burden on gp78-mediated ERAD from misfolded ZAAT reduces Insig turnover, therefore increasing the retention of SCAP-SREBP2 complex in the ER and preventing its transport and activation in the Golgi (Figure 8A). This working model is supported by our observation that suppression of the cholesterol gene expression occurs when ERAD is further compromised in the Z+ *erlec1*^−/−^ liver (Figure 8B). Future investigations targeting gp78 will confirm the involvement of this E3 ligase in mediating ZAAT and Insig1 processing in the zebrafish liver.

The role of ERLEC1 in facilitating the degradation of misfolded antitrypsin was initially studied in cell models expressing NHK, another misfolding, but a non-polymerogenic variant of antitrypsin [51,54,55]. However, the contribution of ERLEC1 in targeting misfolded ZAAT to ERAD has not been previously investigated. To our best knowledge, this is the first study to examine the role of Erlec1 directly in hepatic ZAAT processing in vivo.

To our surprise, ablation of functional Man1b1, which is generally considered to be responsible for the sorting of misfolded proteins in the ER lumen to E3 ligases, does not affect the degradation of misfolded ZAAT, as an increase in intracellular retention and alleviation of ERAD burden was not observed. Our results support the model of protein quality control proposed by Sifers and colleagues: MAN1B1 is a cis-Golgi protein responsible for capturing misfolded proteins that prematurely escape into the secretory pathway and redirecting them back to the ER for refolding or degradation [65,66,67,68]. This explains the increase in ZAAT secretion in the absence of functional Man1b1 proteins in our model without the formation of inclusion bodies: the retrieval of misfolded ZAAT from Golgi to ER is inhibited, thus allowing more ZAAT to be secreted (Figure 8C). Our results also suggest that the effect of kifunensine treatment on ZAAT-expressing human cells which results in increased intracellular retention and secretion may be due to an inhibitory effect on both MAN1B1 and EDEMs. EDEMs, are ER mannosidase-like proteins that are reported to facilitate ERAD targeting [69,70,71,72,73]. It is predicted that the secreted ZAAT in *man1b1a*^−/−^
*man1b1b*^−/−^ fish is functional because ZAAT secreted in kifunensine-treated cells can form inhibitory complexes with its target elastase [31]. The absence of inclusion bodies in these fish suggests that targeting MAN1B1 in vivo to alleviate ZAAT deficiency might increase serum levels without causing liver toxicity.

Apart from cholesterol biosynthesis suppression, this zebrafish model also recapitulates other hepatic responses in existing ZAAT models, such as sensitivity to additional liver stress reported previously [30], and UPR activation appears to be absent based on the bioinformatic analysis in this study. 

Interestingly, UPR activation was reported in a recently established PI*Z transgenic mouse model [74], where expression of human ZAAT is driven by the native hepatocyte- and macrophage-specific promoters of *SERPINA1*. A subset of adult PI*Z mice with high levels of inclusion bodies shows an elevated expression of UPR factors such as XBP1s, BiP, ATF4 and CHOP. However, this is inconsistent with other mouse models where marked up-regulation of CHOP was reported in juvenile PiZ mice but not in adults [75], and up-regulation of other UPR markers such as BiP and XBP-1 was not observed in the inducible Z mice [36]. The reason for the conflicting observations on UPR activation in different mouse models is presently unclear. Overall, there is no compelling evidence that UPR activation is a central response to ZAAT expression in vivo.

In conclusion, our results indicate that the biological processes responding to soluble misfolded ZAAT may be distinguished from those responding to insoluble aggregates through investigations in the zebrafish model, where ZAAT accumulation does not occur. As sequestration in inclusion bodies is unlikely to explain the serum deficiency of ZAAT, the zebrafish model will be useful for examining the processes involved in misfolded protein clearance, which may lead to the development of strategies that can alleviate AATD by enhancing ZAAT secretion without promoting hepatic accumulation.

## 4. Materials and Methods

### 4.1. Plasmids

pCI-neo with either eGFP-T2A-AAT or eGFP-T2A-ZAAT cloned into the *Eco*RI site was described previously [30]. These plasmids were used for the expression of wildtype and Z-mutant antitrypsin in HepG2 cells.

### 4.2. Cell Culturing and Transfection

Human liver cancer cells (HepG2) were maintained at 37 °C with 10% (*v*/*v*) CO_2_ in DMEM medium supplemented with 10% (*v*/*v*) fetal bovine serum, 2 mM glutamine, 50 U/mL penicillin and 50 μg/mL streptomycin. For transfection, cells were seeded into a 24-well tissue culture plate at a density of 8 × 10^4^ cells per well and allowed to incubate at 37 °C overnight for cells to adhere. For each well, a transfection mix composed of 0.5 µg DNA and 1 µL of jetPRIME^®^ transfection reagent (Polyplus, Illkirch, France) was added. At 48 h post-transfection, cells were expanded into a 6-well tissue culture plate for G418 selection. 

### 4.3. Reverse Transcriptase PCR

Total RNA was isolated from cells or fish liver using Trizol lysis reagent (Sigma Aldrich, St. Louis, MO, USA) followed by isopropanol precipitation. The RNA pellet was washed with 70% (*v*/*v*) ethanol. RNA was quantitated using a nanodrop spectrophotometer and cDNA synthesis was performed using MMLV Reverse Transcriptase (MMLV RT [H-], Promega, Madison, WI, USA) and oligo-dT primer. For the detection of cholesterol biosynthesis pathway genes, primers listed in Appendix A were used. PCR was performed using 2X GoTaq^®^ Green Master Mix (Promega).

### 4.4. Real-Time Quantitative PCR

cDNA and primers were added to the LightCycler^®^ 480 SYBR Green I Master mix (Roche, Basel, Switzerland) for the qPCR reaction using the LightCycler^®^ 480 Instrument with the following cycling conditions: 95 °C for 10 s denaturation, 58 °C for 30 s annealing, 72 °C for 30 s extension, for 45 cycles. Expression levels of transcripts were normalised to at least two housekeeping genes as indicated for each experiment and were calculated using the following method:
ΔCt = Average Ct _(target gene)_ − Average Geomean Ct _(housekeeping genes)_.(1)
Relative expression level = 2^−ΔCt^ × 10,000 (constant).

### 4.5. Zebrafish Maintenance

Zebrafish were maintained at 28 °C on a 14 h light/10 h dark cycle. Embryos were maintained in a 10 cm petri dish in E3 buffer in the 28 °C incubator up until 5 days post fertilisation (dpf). Transgenic lines expressing human α1-antitrypsin were maintained as heterozygous adults and were outcrossed to wildtype TU fish to generate transgenic fish and non-transgenic siblings for analysis. All animals were handled according to standard operating procedures and approved by the Monash University Animal Ethics Research Committee (ERM14549, approved 26 June 2018; ERM14732, approved 8 October 2018; and ERM30408, approved 25 January 2022).

### 4.6. Mutant Zebrafish Lines

A small volume (~2 nL) of injection mix containing 1 M KCl, 30 µM CRISPR gRNA, 20 µM Cas9 protein (Integrated DNA Technologies, Coralville, IA, USA), 0.5% cascade blue and 0.5% phenol red was injected into each one-cell embryo using the Femtojet microinjector (Eppendorf, Hamburg, Germany) under a dissecting microscope. The injected embryos were transferred to a petri dish and maintained at 28 °C. At 5 dpf, DNA was extracted from a tail biopsy of each embryo and PCR amplification across the CRISPR-target site was performed to indicate successful CRISPR/Cas9-induced mutations. The PCR product was separated via non-denaturing PAGE. Embryos with induced mutations were raised to adulthood when founder(s) could be identified.

### 4.7. Liquid Chromatography-Mass Spectrometry (LC-MS/MS)

Adult zebrafish livers were solubilised in 1% sodium deoxycholate (SDC) using standard protocols. Briefly, extracted livers were placed in 1% SDC in 100 mM Tris pH 8.1 and homogenised for 3 × 10 s using an electric homogeniser. Samples were boiled at 95 °C for 5 min and then sonicated for 3 × 10 s (samples were cooled on ice between sonication bursts). Samples were fractionated by centrifugation at 15,000*g* for 5 min. The soluble fraction was collected, and the protein concentration was measured. Up to 400 µg of liver protein was digested with trypsin for ~12 h on a column using standard filter-aided sample preparation (FASP) manufacturer protocol (Expedeon, Cambridge, UK). Digested protein was concentrated to 50 µL using a Speedvac, cleaned up using ZipTip (following the manufacturer’s protocol) and concentrated again to ~5 µL. Samples were diluted to 20 µL in 0.1% formic acid and iRT peptides were added for mass-spectrometry analysis.

Mass spectrometry was performed by the Monash Proteomics & Metabolomics Facility. Spectral data were acquired using a Dionex Ultimate 3000 RSLCnano system coupled to an Orbitrap Fusion Tribrid (Thermo Scientific, Waltham, MA, USA) mass Spectrometer. An acclaim PepMap RSLC (75 μm × 50 cm, nanoViper, C18, 2 μm, 100 Å; Thermo Scientific) analytical column and an Acclaim PepMap 100 (100 μm × 2 cm, nanoViper, C18, 5 μm, 100 Å; Thermo Scientific) trap column was used to chromatographically separate the tryptic peptides. Mass spectrometric DDA (data-dependent acquisition) data were searched against a modified version of the Uniprote Zebrafish database (GFP, human AAT, iRT peptides added) using Byonic (ProteinMetrics, Cupertino, CA, USA) embedded into Proteome Discoverer (Thermo Scientific) considering a 1% false discovery rate (FDR). Spectronaut 9 was used for both the generation of a sample-specific spectral library as well as to interrogate all mass spectrometric DIA files.

### 4.8. RNA-Sequencing

Extracted adult zebrafish livers were solubilised in Trizol lysis reagent (Sigma Aldrich) and total RNA was isolated by chloroform and isopropanol precipitation following the manufacturer’s protocol. RNA sequencing was performed by Monash Health Translation Precinct (MHTP) Medical Genomics Facility (Clayton, Australia). Briefly, samples were analysed using a QC bioanalyzer having an average RIN of 8.8 (7–10) and an average concentration of 290 ng/µL using Qubit. 400 ng of RNA per sample was used for sequencing. Samples were sequenced over two runs using an Illumina NextSeq550 using 75 bp paired-end sequencing (High output mode V2 chemistry 1% PhiX). 2.3 pM of libraries (~340 bp in length) were loaded onto the flow cell for optimal clustering, with >88% passing the clustering filter and used for subsequent sequencing detection. A total of 1.27 billion reads were detected.

The RNAseq data were aligned to a custom reference genome using the RNAsik pipeline (version 1.4.7) [40]. The custom genome was generated by concatenating the fasta files from the GRCz11 Ensembl reference (release 92) and the AAT transgene. The annotation GTF file for GRCz11 was also altered to contain a corresponding entry for the transgene. Both the custom fasta file and the gtf file were then filtered to remove contigs as well as unknown and alternate chromosomes. The RNAsik pipeline was run with the following parameters: ‘-align star -paired -all’ and used the combined fasta file and gtf file as input for the ‘-fastaRef’ and ‘-gtfFile’ parameters. In brief, the RNAsik pipeline used STAR (v2.5.2b) [39] to align the raw reads to the reference genome and featureCounts to count high-quality aligned reads to annotated genes. The resulting count matrix was then used for differential gene expression analysis. 

### 4.9. Differential Gene/Protein Analysis

Transcript counts from approximately 31,400 features were consolidated to corresponding unique genes (approximately 28,000). Counts were then converted to reads per kilobase of the transcript, per million mapped reads (RPKM). For differential expression analysis, lowly expressed genes are excluded for better estimation of the mean-variance relationship. Only transcripts and proteins with raw counts above 100 in at least 3 samples were considered to distinguish expression from noise. 

DEGs/DEPs were defined as genes/proteins with significant dysregulation in the ZAAT-expressing liver samples only. Multiple Student’s *t*-test analysis was used to compare the log₂RPKM or protein counts for the following genotypes: (1) Z+ vs. A+; (2) Z+ vs. [Z- & A-]; (3) Z+ vs. Z-; and (4) Z- vs. A- with summed false discovery rate (FDR) threshold applied (for statistical parameters applied, see Appendix A). 

### 4.10. Antibodies

The following antibodies were purchased: rabbit anti-human α1-antitrypsin (Sigma Aldrich #A0409), used at 1:1000 for immunoblots; Mouse anti-GFP (clones 7.1 and 13.1) (Roche #11814460001), used at 1:1000 for immunoblots; Mouse anti-β-tubulin (AA2) (Millipore #05-661), used at 1:5000 for immunoblots; Sheep anti-Rabbit Ig-HRP conjugated (pre-adsorbed) (Rockland Immunochemicals, Pottstown, PA, USA) used at 1:5000 for immunoblots; Goat anti-Mouse Ig-HRP conjugated (pre-adsorbed) (Rockland Immunochemicals) used at 1:5000 for immunoblots. The rabbit anti-zebrafish IgM antibody (used at 1:2000 for immunoblots) was obtained from Dr J. Coll.

### 4.11. Protein Lysates and Immunoblotting

Proteins were solubilised in Laemmli sample buffer (62.5 mM Tris-HCl, pH 6.8, 2% (*w/v*) SDS, 10% (*v/v*) glycerol, 0.01% (*w/v*) bromophenol) with 0.05 M 2-mercaptoethanol. Samples were boiled and resolved in 10% acrylamide gels and transferred onto PVDF membranes. Membranes were incubated in 5% (*w/v*) skim milk in TBS buffer for at least 1 h at room temperature, then in primary antibodies diluted appropriately in TBS overnight at 4 °C. Membranes were washed with TBS-T for 3 × 15 min and incubated with HRP-conjugated secondary antibody diluted appropriately in TBS for 1 h at room temperature. After washing with TBS-T for 3 × 15 min, membranes were incubated in enhanced chemiluminescence (ECL) developing solution (Western Lightning Plus, PerkinElmer, Waltham, MA, USA) for approximately 30 s. The membrane was exposed to Fuji Super RX X-ray films. 

Membranes were washed once with TBS-T and incubated with stripping buffer (1.5% (*w/v*) glycine, 0.1% (*w/v*) SDS, 1% (*v/v*) Tween 20, pH 2.2) for 2 × 10 min at room temperature followed by 2 × 10 min washes with PBS then 2 × 10 min washes with TBS-T. Membranes were incubated for 1 h at room temperature with rocking, in blocking solution before re-probing or staining with 0.008% (*w/v*) Direct blue 71 in 40% (*v/v*) ethanol and 10% (*v/v*) glacial acetic acid for total protein.

### 4.12. Total Cholesterol Assay

Adult zebrafish were starved for 24 h prior to euthanasia. The liver was harvested and hepatic cholesterol levels were measured using an enzymatic kit (BioVision #K623-100) following the manufacturer’s instructions. The liver pellet post lipid extraction was dried at 50 °C and pellet mass was measured using an analytical balance.

### 4.13. Periodic Acid-Schiff (PAS) Diastase (D) Staining

All histological work was performed by the Monash Histology Platform. Paraffin-embedded zebrafish liver was sectioned at a thickness of 3 µm and stained for PAS (with or without diastase) as previously described [30]. Sections were scanned using the Aperio ScanScope^®^ AT Turbo for visualisation.

## Figures and Tables

**Figure 1 ijms-24-02475-f001:**
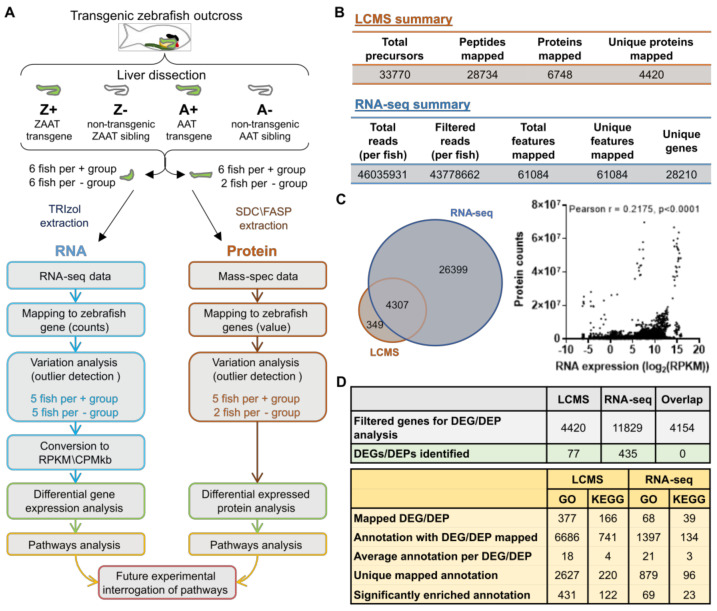
Overview of the global transcriptomic/proteomic approach to identify dysregulated pathways in ZAAT-expressing zebrafish livers. (**A**) Six fish in each genotype group were analysed, with protein and RNA samples extracted from each liver sample. Fish identified as outliers were excluded from subsequent analysis (see Appendix A). (**B**) Summary statistics of LC-MS/MS and RNA-seq data aligned to the reference library or genome. (**C**) Comparison of libraries generated from RNA-seq and LC-MS/MS data (left panel). Correlation analysis of transcriptomic and proteomic expression profiles (right panel). Each point represents protein (counts) and RNA (log_2_CPM/kb) expression data from one gene in one fish. (Pearson r, *r* = 0.2157, *p* < 0.0001). (**D**) Summary statistics of differential expression analysis (top panel) and pathway enrichment analysis (bottom panel).

**Figure 2 ijms-24-02475-f002:**
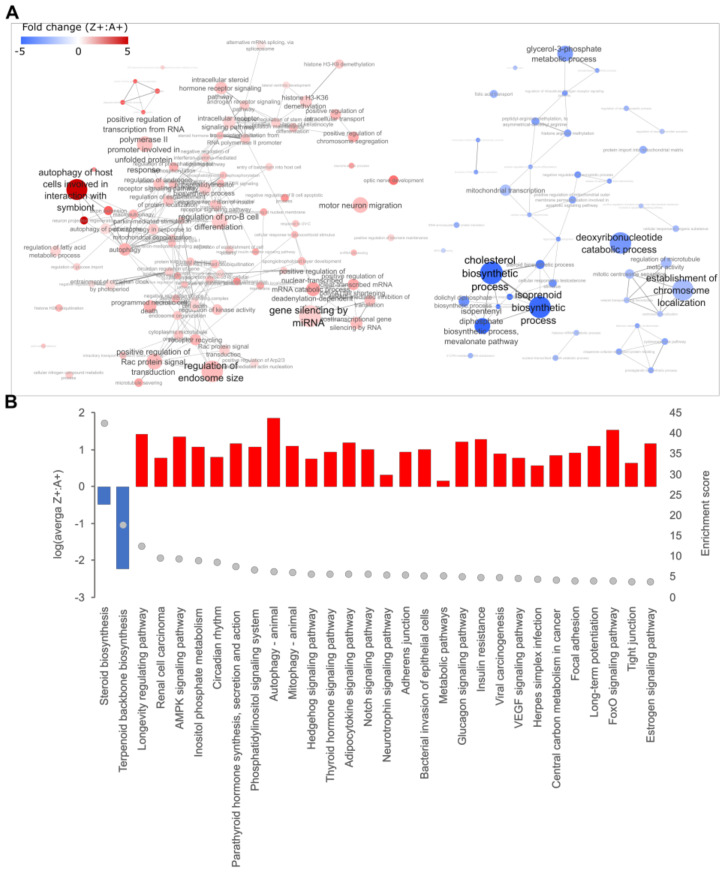
Pathway enrichment analysis of DEGs identified from RNA-seq data. Differentially expressed genes (DEGs) from RNA-seq data were mapped to the corresponding human orthologs and (**A**) Gene Ontology (GO) enrichment analysis was performed. The enriched GO annotations under the ‘Biological Process’ category were filtered based on an enrichment threshold >4 with annotations containing DEGs that are either all up- or down-regulated. Filtered annotations were then mapped onto an interaction network as nodes and were connected by an edge if there were common DEGs present between nodes. The thickness of the edge correlates to the number of common DEGs shared between two nodes. The size of each node represents the enrichment score from the GO enrichment analysis. Red indicates the up-regulation of the GO term while blue indicates downregulation. (**B**) The top 30 results from KEGG enrichment analysis. The histogram shows the direction and average level of dysregulation for all DEGs mapped to each KEGG pathway (left *y*-axis), and the grey dots indicate enrichment levels (right *y*-axis).

**Figure 3 ijms-24-02475-f003:**
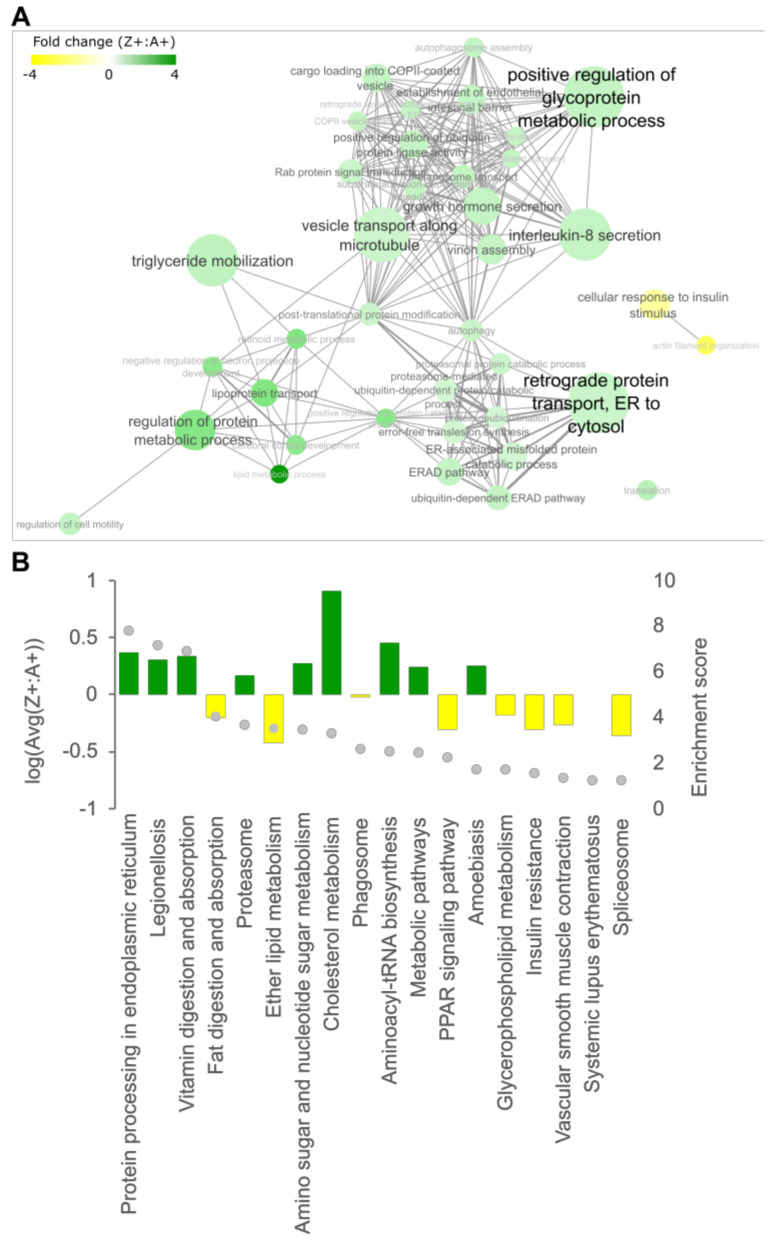
Pathway enrichment analysis of DEPs identified from LC-MS/MS data. Differentially expressed proteins (DEPs) from LC-MS/MS data were mapped to the corresponding human orthologs and (**A**) Gene Ontology (GO) enrichment analysis was performed. The enriched GO annotations under the ‘Biological Process’ category were filtered for annotations containing DEPs that are either all up- or down-regulated. Filtered annotations were then mapped onto an interaction network as nodes and were connected by an edge if there are common DEPs present between nodes. The thickness of the edge correlates to the number of common DEPs shared between two nodes. The size of each node represents the enrichment score from the GO enrichment analysis. Green indicates the up-regulation of the GO term while yellow indicates downregulation. (**B**) KEGG enrichment analysis of DEPs. The histogram shows the direction and average level of dysregulation for all DEPs mapped to each KEGG pathway (left *y*-axis), and the grey dots indicate enrichment levels (right *y*-axis).

**Figure 4 ijms-24-02475-f004:**
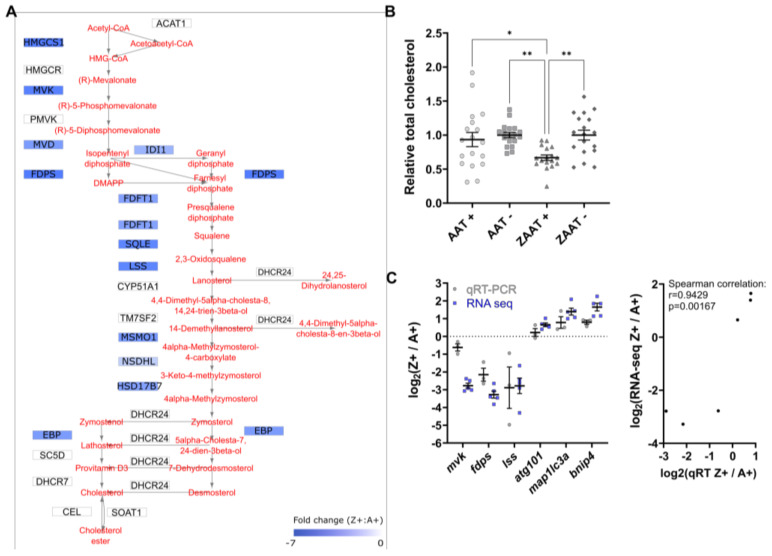
Cholesterol biosynthesis is suppressed in ZAAT-expressing zebrafish liver. (**A**) Mapping of DEGs onto KEGG pathways suggests downregulation of cholesterol biosynthesis (modified from pathways hsa00900 and hsa00100). (**B**) Total cholesterol was measured in adult zebrafish livers. A total of 18 fish from each genotype, pooled from 5 independent experiments, were used for analysis. Every single point represents the average of two technical replicate measurements for each animal. One-way ANOVA was used to determine statistical significance. Error bars indicate mean ± s.e.m. * *p* < 0.05, ** *p* < 0.01. (**C**) Validation of RNA-seq by qPCR. Expression profiles of six DEGs from RNA-seq and qPCR data were compared (left panel). qPCR expression values were normalised to the housekeeping genes *rpl13a* and *lsm12b*. Correlation analysis of RNA-seq and qPCR (right panel), Spearman’s rho, *r*_s_ = 0.9429, *p* = 0.00167.

**Figure 5 ijms-24-02475-f005:**
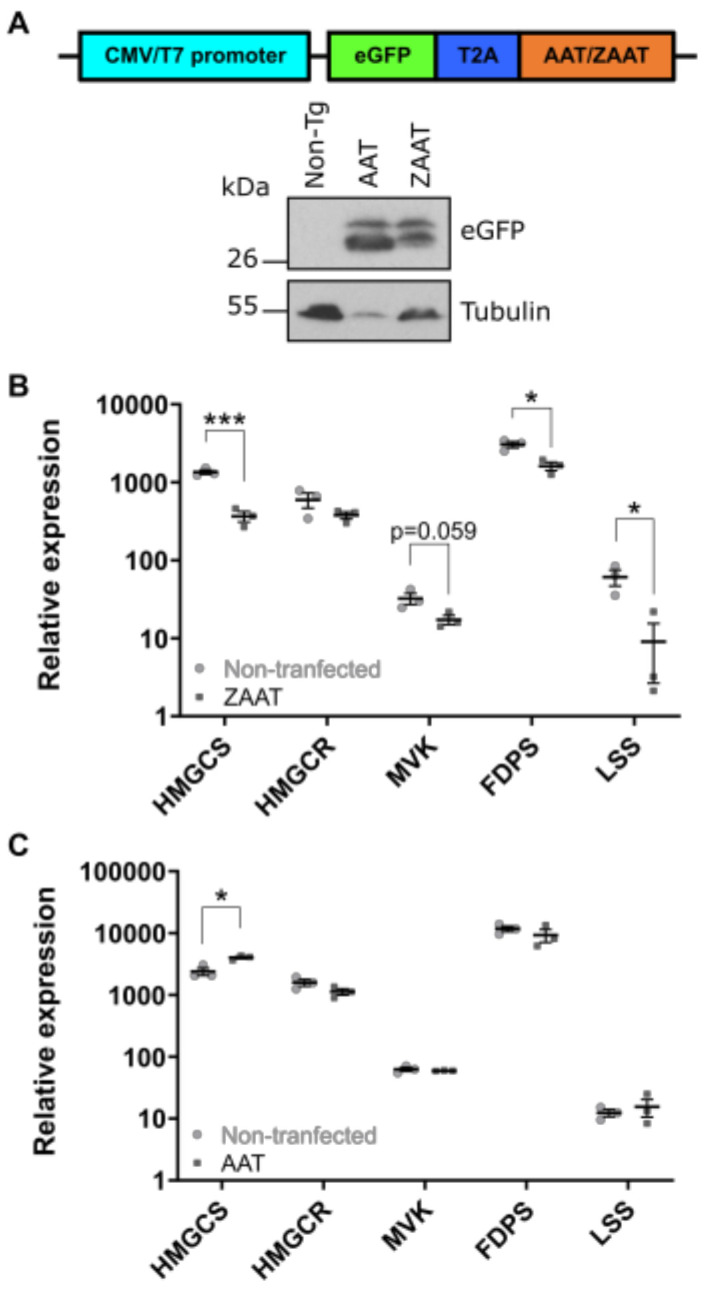
Expression of ZAAT in HepG2 cells suppresses SREBP2 target genes. (**A**) Transgenes used for HepG2 transfection (top panel). Immunoblot of HepG2 cell lysates for eGFP as an indication of transgene expression levels using β-tubulin as a loading control (bottom panel). qPCR analysis was performed from cDNA generated from HepG2 cells with stable overexpression of (**B**) eGFP-T2A-ZAAT and (**C**) eGFP-T2A-AAT. Expression values were normalised to the housekeeping genes *GAPDH*, *ACTB* and *TBP*. An unpaired student’s *t*-test was used to determine the statistical significance for each gene. Error bars indicate mean ± s.e.m. * *p* < 0.05, *** *p* < 0.001.

**Figure 6 ijms-24-02475-f006:**
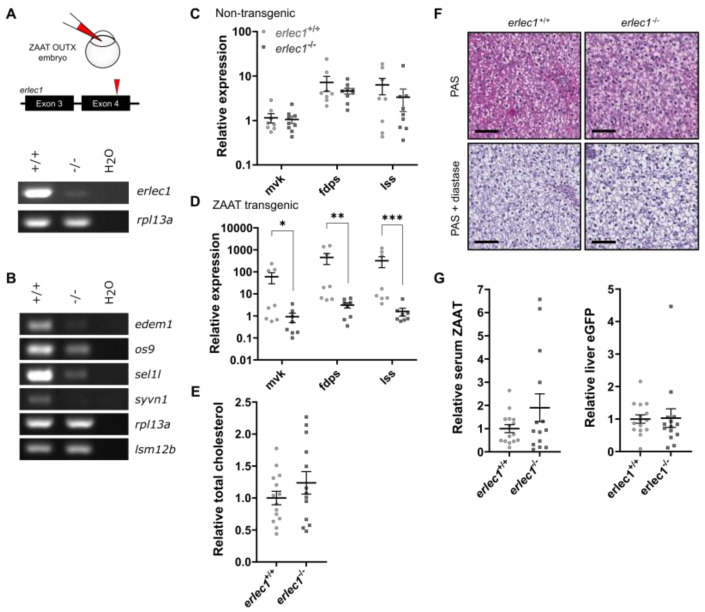
Erlec1 ablation further suppresses cholesterol biosynthesis in ZAAT-expressing zebrafish liver. (**A**) gRNA targeting exon 4 was injected into embryos from the Tg(*lfabp*: eGFP-T2A-ZAAT) line at the one-cell stage. RT-PCR for *erlec1* used cDNA made from 1 µg RNA extracted from pooled whole embryos at 7 dpf. The housekeeping gene *rpl13a* was used as a loading control. (**B**) RT-PCR of genes encoding components of Hrd1 E3 ligase complex using cDNA made from 1 µg RNA extracted from adult zebrafish liver. The housekeeping genes *rpl13a* and *lsm12b* were used as loading controls. qPCR analysis of representative cholesterol biosynthesis DEGs in adult (**C**) non-transgenic zebrafish livers, as well as (**D**) livers expressing ZAAT, using data pooled from two independent experiments. Expression values were normalised to the housekeeping genes *fabp10a*, *rpl13a,* and *lsm12b*. Error bars indicate mean ± s.e.m. Unpaired student’s *t*-test and Mann-Whitney U test were used to detect statistical significance for each gene. * *p* < 0.05, ** *p* < 0.01, *** *p* < 0.001. (**E**) Total cholesterol was measured in adult ZAAT-expressing livers (*erlec1*^+/+^, *n* = 14, *erlec1*^−/−^, *n* = 13). Every single point represents the average of two technical replicate measurements for each animal. Student’s *t*-test was performed and indicated no significant difference in levels from wildtype and *erlec1*^−/−^ fish. (**F**) Periodic acid-Schiff (PAS) staining with and without diastase was performed on 3 µm thick paraffin zebrafish liver sections. The scale bar represents 50 µm. (**G**) The ratio of antitrypsin to IgM (left panel) and eGFP to total protein (right panel) in zebrafish blood was detected by immunoblot (see Appendix A) and quantitated by densitometry. Shown are the pooled data from four independent experiments, with each point representing an individual fish. Student’s *t*-test was performed and indicated no significant difference between levels in wildtype and *erlec1*^−/−^ fish.

**Figure 7 ijms-24-02475-f007:**
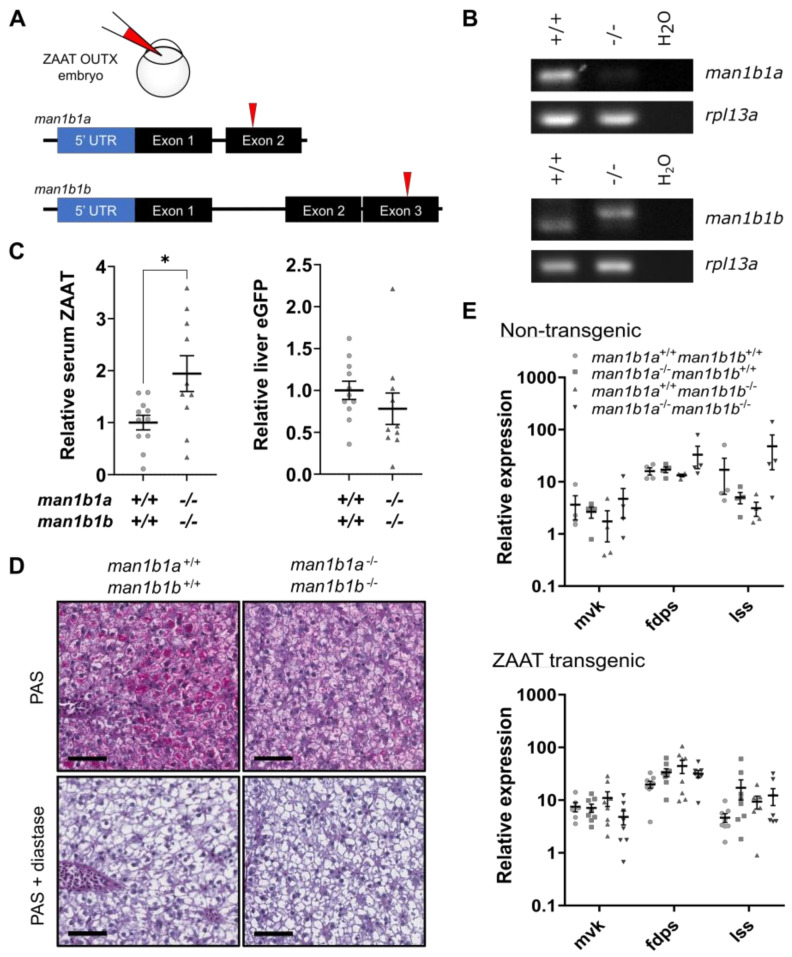
Man1b1 ablation enhances ZAAT secretion without affecting retention in hepatocytes. (**A**) Two gRNA targeting exon 2 of *man1b1a* or exon 3 of *man1b1b* were co-injected into outcross embryos from the Tg(*lfabp*: eGFP-T2A-ZAAT) line at the one-cell stage. (**B**) RT-PCR of *man1b1a* (top panel) and *man1b1b* (bottom panel) using cDNA made from 1 µg RNA extracted from pooled whole embryos at 7 dpf. The housekeeping gene *rpl13a* was used as a loading control. (**C**) The ratio of antitrypsin to IgM (left panel) and eGFP to total protein (right panel) in zebrafish blood was detected by immunoblot (Appendix A) and quantitated by densitometry. Shown are the pooled data from three independent experiments, with each point representing an individual fish. Statistical significance was determined by Unpaired Student’s *t*-test. * *p* < 0.05. (**D**) Periodic acid-Schiff (PAS) staining with and without diastase was performed on 3 µm thick paraffin zebrafish liver sections. The scale bar represents 50 µm. (**E**) qPCR analysis of representative cholesterol biosynthesis DEGs in adult non-transgenic zebrafish livers (top panel), as well as livers expressing ZAAT (bottom panel), using data pooled from two independent experiments. Expression values were normalised to the housekeeping genes *rpl13a* and *lsm12b*. One-way ANOVA was performed and indicated no significant difference between levels from the four genotypes for each gene.

**Figure 8 ijms-24-02475-f008:**
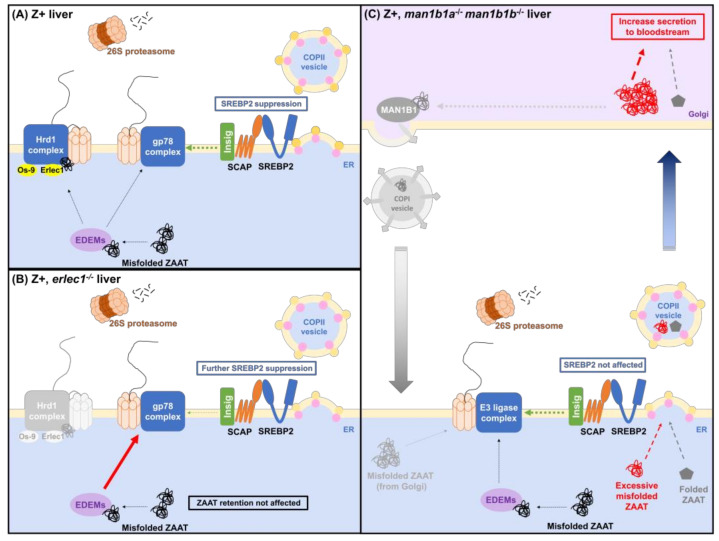
Model for ERAD pathway interference between the ZAAT processing system and cholesterol biosynthesis. (**A**) In zebrafish expressing ZAAT, degradation of misfolded ZAAT via ERAD-L limits the ERAD machinery available for Insig turnover via ERAD-M. More Insig, therefore, remains bound to the Scap-Srebp2 complex in the ER membrane, reducing Srebp2 release from the ER and suppressing activation of the downstream cholesterol biosynthesis pathway. (**B**) Ablation of Erlec1 prevents the retrotranslocation of misfolded ZAAT via the Hrd1 ligase complex and redirects it to the gp78 complex, further reducing the capacity for Insig degradation and enhancing Srebp2 retention in the ER membrane. (**C**) The absence of Man1b1 proteins in zebrafish liver enhances ZAAT secretion, consistent with a role for Man1b1 in the Golgi in retrieving misfolded ZAAT that escaped the ERQC system and returning it to the ER.

## Data Availability

The mass spectrometry proteomics data have been deposited to the ProteomeXchange Consortium via the PRIDE [76] partner repository with the dataset identifier PXD036830. The RNAseq data have been deposited in NCBI’s Gene Expression Omnibus [77] and are accessible through GEO Series accession number GSE215899. All other data files are available at https://figshare.com/s/a7fe6149482d21aabf3a, DOI: 10.26180/21114691.

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
