# Peer review of "Expression of the Z Variant of α1-Antitrypsin Suppresses Hepatic Cholesterol Biosynthesis in Transgenic Zebrafish"

_ijms, 2023, doi:10.3390/ijms24032475_

Round 1
Reviewer 1 Report
Journal IJMS (ISSN 1422-0067)
Manuscript ID ijms-2172606
Type Article
Title Expression of the Z variant of α1-antitrypsin suppresses hepatic cholesterol biosynthesis in transgenic zebrafish
Authors Connie Fung, Brendan Wilding , Ralf B. Schittenhelm , Robert J. Bryson-Richardson , Phillip I. Bird
Section Molecular Pathology, Diagnostics, and Therapeutics
Special Issue Zebrafish as a Model in Human Disase
Comment
This is a well-designed functional work reporting proteomic, transcriptomic and biochemical data obtained in vitro and in vivo with a CRISPR/Cas9 gene editing transgenic zebrafish model on the role of Pi*Z allele of SERPINA1 (ZAAT) in the context of hepatic cholesterol biosynthesis pathway. The work is well written and well organized, while the experimental design is well performed, too. This study will undoubtedly increase our knowledge on the implication of α1-antitrypsin deficiency in humans and I therefore suggest its publication on IJMS. Please see below a few minor points:
1. First lines, besides the well-known inactivating mutations, rare deficiency alleles have also been identified (PMID: 27296815) In addition, the expression of α1-antitrypsin is also under epigenetic control (PMID: 33015055). Please include both notions.
2. More supporting references should be included in the methods
3. Please include a sentence on the role of α1-antitrypsin deficiency in the etiology of chronic respiratory disorders PMID: 32051168
4. Please remove spaces between paragraphs
5. For a better reading, HepG2 should be described as liver cancer cells in the methods. Same comment for HEK293 cells mentioned in line 325
Author Response
Review 1:
This is a well-designed functional work reporting proteomic, transcriptomic and biochemical data obtained in vitro and in vivo with a CRISPR/Cas9 gene editing transgenic zebrafish model on the role of Pi*Z allele of SERPINA1 (ZAAT) in the context of hepatic cholesterol biosynthesis pathway. The work is well written and well organized, while the experimental design is well performed, too. This study will undoubtedly increase our knowledge on the implication of α1-antitrypsin deficiency in humans and I therefore suggest its publication on IJMS. Please see below a few minor points:
- First lines, besides the well-known inactivating mutations,rare deficiency alleles have also been identified (PMID: 27296815) In addition, the expression of α1-antitrypsin is also under epigenetic control (PMID: 33015055). Please include both notions.
Response: a sentence outlining null alleles (including the rare deficiency alleles reported in the suggested reference PMID: 27296815) is now included in lines 30-32.
The reference PMID: 33015055 shows that the expression of α1-antitrypsin can be regulated epigenetically in human PMBCs, but did not extend its findings to hepatocytes. Epigenetic regulation of SERPINA1 would be useful in studies of AATD where expression of the SERPINA1 gene is affected (such as null alleles). As our zebrafish model investigated AATD caused by a protein misfolding variant, this reference provided by the reviewer may not be relevant. Although epigenetic changes have been reported in the PiZZ liver that may be associated with disease progression [PMID: 30681738], investigation of the epigenome remains outside the scope of the present study.
- More supporting references should be included in the methods
Response: references for bioinformatic software used for analysis were included in both the main text and the methods section. Sufficient descriptions were provided for the remaining analysis/protocols including the source of reagents and experimental conditions, these were optimised based on in-house laboratory protocols or instructions from reagent manufacturers, therefore further referencing may not be necessary.
- Please include a sentence on the role of α1-antitrypsin deficiency in the etiology of chronic respiratory disorders PMID: 32051168
Response: sentence and reference are now included in lines 28-29 as suggested.
- Please remove spaces between paragraphs
Response: spaces between paragraphs are now removed as suggested.
- For a better reading, HepG2 should be described as liver cancer cells in the methods. Same comment for HEK293 cells mentioned in line 325
Response: cell line descriptions are now included in lines 595 and 314 as suggested.
Reviewer 2 Report
This manuscript describes original results showing the effect of human Z-AAT in transgenic zebrafish models, which was previously developed by the same group. The authors used RNA-seq and proteomic analyses and showed that ZAAT suppressed cholesterol biosynthesis. These results are off the general interest in the field. The experiments were well designed and performed. The authors may consider the following points:
Fig S1, the liver sections and the antibody used should be clearly label in the Fig and Fig legend. For example, AAT may be liver section from AAT tg zebrafish; alpha 1 antitrypsin may be antibody against human alpha 1 antitrypsin. Positive control (e.g. ZAAT human liver) should be included in the experiment.
Fig 1, authors may clarify what is the difference between Z- and A- groups. Are they genetically the same? After identify outliers, n= for each group should be clearly indicated.
In lines 177-120, the authors stated “For the RNA-seq data set, 377 DEGs (86.7%) were successfully 117 mapped to GO terms, and 166 genes (38.2%) were mapped to KEGG pathways. For the 118 LC-MS/MS data set, 65 (84.4%) and 38 (49.4%) DEPs were mapped to GO and KEGG path-119 ways respectively (Figure 1D)”. However, Fig 1D did not show these data.
Table S1 and S2 should also show data of Z+: Z- and A+:A- for genes and proteins. These results are also important to show the effect of wild type human AAT as well as ZAAT in zebrafish.
The authors may also discuss the effect of wild type AAT in zebrafish as the data has been generated.
Section 2.3 and 2.4 may be written in more straightforward minor. Some of the information can removed or move to discussion section.
Fig 8 may be simplified and more focused on the key factors.
Author Response
Reviewer 2:
This manuscript describes original results showing the effect of human Z-AAT in transgenic zebrafish models, which was previously developed by the same group. The authors used RNA-seq and proteomic analyses and showed that ZAAT suppressed cholesterol biosynthesis. These results are off the general interest in the field. The experiments were well designed and performed. The authors may consider the following points:
- Fig S1, the liver sections and the antibody used should be clearly label in the Fig and Fig legend. For example, AAT may be liver section from AAT tg zebrafish; alpha 1 antitrypsin may be antibody against human alpha 1 antitrypsin. Positive control (e.g. ZAAT human liver) should be included in the experiment.
Response: The figure legend for S1 is revised as suggested. A negative control (non-transgenic zebrafish liver) is included to show that the staining observed is specific to human antitrypsin in the zebrafish liver. We followed the same procedure from our previous study (reference #30) where staining of liver sections from healthy individuals and PiZZ patients indicated this experimental procedure is successful to detect ZAAT inclusions if present. Unfortunately, these tissues were no longer available to us to perform the staining in this study.
- Fig 1, authors may clarify what is the difference between Z- and A- groups. Are they genetically the same? After identifying outliers, n= for each group should be clearly indicated.
Response: the Z- and A- groups are genetically different as they are derived from two independent transgenic lines described in reference #30. The two independent lines are maintained as heterozygous adults and were outcrossed to wildtype fish to generate the four genotypes (outlined in lines 78-80) used for analysis. This information is now included in the methods section (lines 624-626).
The number of animals used in each group after outlier identification is now indicated in Figure 1A.
- In lines 177-120, the authors stated “For the RNA-seq data set, 377 DEGs (86.7%) were successfully mapped to GO terms, and 166 genes (38.2%) were mapped to KEGG pathways. For the LC-MS/MS data set, 65 (84.4%) and 38 (49.4%) DEPs were mapped to GO and KEGG pathways respectively (Figure 1D)”. However, Fig 1D did not show these data.
Response: The numbers in lines 117-120 refer to the number of DEG/DEP that was mapped to one or more annotations. The numbers in Figure 1D refers to the number of GO/KEGG annotation with one or more DEG/DEP mapped and obtained an enrichment score of >1.
Additional summary statistics from each step of the functional annotation and pathway enrichment analysis are now included in Figure 1D for clarification (with minor changes to the numbers in line 114 to be consistent with the updated analysis).
- Table S1 and S2 should also show data of Z+: Z- and A+:A- for genes and proteins. These results are also important to show the effect of wildtype human AAT as well as ZAAT in zebrafish.
Response: data of A+/A- and Z+/Z- for DEs and DEPs are now included in Table S1 and S2.
- The authors may also discuss the effect of wildtype AAT in zebrafish as the data has been generated.
Response: Our previous study (reference #30) shows that AAT is being synthesized and processed in a similar manner in zebrafish as in mammalian systems without any detrimental effects. While the effect of wildtype AAT on the zebrafish liver proteome/transcriptome is outside the scope of this study, it would be interesting to pursue this in the future to further our understanding of the hepatic proteostasis network in zebrafish when overexpressing human proteins.
- Section 2.3 and 2.4 may be written in more straightforward minor. Some of the information can removed or move to discussion section.
Response: Sufficient information regarding different components of ERAD and cholesterol regulation is necessary to explain our working hypothesis, which we developed in an attempt to explain our bioinformatic observations. This information clarifies the rationale behind our choice to manipulate the erlec1, man1b1a and man1b1b genes, and the choice of analysis for the mutant lines generated and would be useful to the readers before presenting the analysis data in these sections.
Section 2.3 is now revised to introduce the regulation of cholesterol biosynthesis in a more concise manner.
- Fig 8 may be simplified and more focused on the key factors.
Response: the diagrams in Figure 8 are now simplified as suggested.